# The distribution of fitness effects among synonymous mutations in a gene under directional selection

Eleonore Lebeuf-Taylor[1†], Nick McCloskey[1†], Susan F Bailey[2], Aaron Hinz[1], Rees Kassen[1]*

[1]Department of Biology, University of Ottawa, Ottawa, Canada; [2]Department of Biology, Clarkson University, Potsdam, United States

**Abstract** The fitness effects of synonymous mutations, nucleotide changes that do not alter the encoded amino acid, have often been assumed to be neutral, but a growing body of evidence suggests otherwise. We used site-directed mutagenesis coupled with direct measures of competitive fitness to estimate the distribution of fitness effects among synonymous mutations for a gene under directional selection and capable of adapting via synonymous nucleotide changes. Synonymous mutations had highly variable fitness effects, both deleterious and beneficial, resembling those of nonsynonymous mutations in the same gene. This variation in fitness was underlain by changes in transcription linked to the creation of internal promoter sites. A positive correlation between fitness and the presence of synonymous substitutions across a phylogeny of related Pseudomonads suggests these mutations may be common in nature. Taken together, our results provide the most compelling evidence to date that synonymous mutations with non-neutral fitness effects may in fact be commonplace.
DOI: https://doi.org/10.7554/eLife.45952.001

*For correspondence:
rees.kassen@uottawa.ca

†These authors contributed equally to this work

Competing interests: The authors declare that no competing interests exist.

## Introduction

Our ability to use DNA sequence data to make inferences about the evolutionary process from genes or genomes often relies on the assumption that synonymous mutations, those that do not result in an amino acid change, are neutral with respect to fitness. Yet there is compelling evidence that this assumption is sometimes wrong: comparative (*Lawrie et al., 2013*) and experimental (*Lind et al., 2010*) data show that synonymous mutations can have a range of fitness effects from negative to positive, and can even contribute to adaptation (*Bailey et al., 2014*; *Agashe et al., 2016*; *Kristofich et al., 2018*; *She and Jarosz, 2018*). A range of mechanisms including codon usage bias, altered mRNA structure, and the creation of promoter sequences could lead to changes in the rate or efficiency of transcription, translation, and/or protein folding and/or expression that, in turn, impact fitness (*Plotkin and Kudla, 2011*). The specific mechanism notwithstanding, it is clear that synonymous mutations are not always neutral; however, the degree of variability in their fitness effects, and how often they contribute to adaptation, remains unknown.

Our work focuses on testing fitness effects of synonymous mutations in a gene known to be under selection. Previous work by *Bailey et al. (2014)* reported the discovery of two spontaneous, highly beneficial synonymous mutations arising independently over the course of a selection experiment in *gtsB*, a gene that codes for a membrane-bound permease subunit of an ABC glucose-transporter in the Gram-negative bacterium *Pseudomonas fluorescens* SBW25. The *gts* operon is crucial to glucose uptake, as it encodes a four-protein system that binds glucose in the periplasm and actively transports it across the inner membrane. Knockouts of *gtsB* show that this particular gene, the second in

the operon, is targeted by selection in an environment where low glucose limits growth (*Bailey et al., 2014*).

## Results

As a first step towards understanding the evolutionary effects of synonymous mutations, we estimated the distribution of fitness effects (DFE) for 39 synonymous, 65 nonsynonymous, and six nonsense substitutions at 34 sites along *gtsB*. Single nucleotide mutants were generated through site-directed mutagenesis and competed against the ancestor strain in glucose-limited medium. We previously reported on two beneficial synonymous mutations in this gene recovered from a population that had evolved for ~1000 generations in glucose-limited medium and confirmed that *gtsB* is a target of selection under these conditions (*Bailey et al., 2014*). Visual inspection of the DFEs for nonsynonymous and synonymous mutations (*Figure 1A*) reveals they are similar, with both having modes close to neutrality ($w \sim 1$) and substantial variation that includes mutants with both positive and deleterious effects. However, the distributions differ significantly (p=0.0002 based on a bootstrapped estimate of the Kolmogorov-Smirnov D-statistic from 10,000 permutations) due to the presence of a handful of strongly deleterious nonsense mutations in the non-synonymous set that presumably produce a truncated, non-functional protein.

Remarkably, the DFEs for beneficial nonsynonymous and synonymous mutations are indistinguishable (bootstrapped K-S test, p=0.59), suggesting that both kinds of mutation could contribute to adaptation. The combined DFE for both kinds of beneficial mutations is approximately L-shaped,

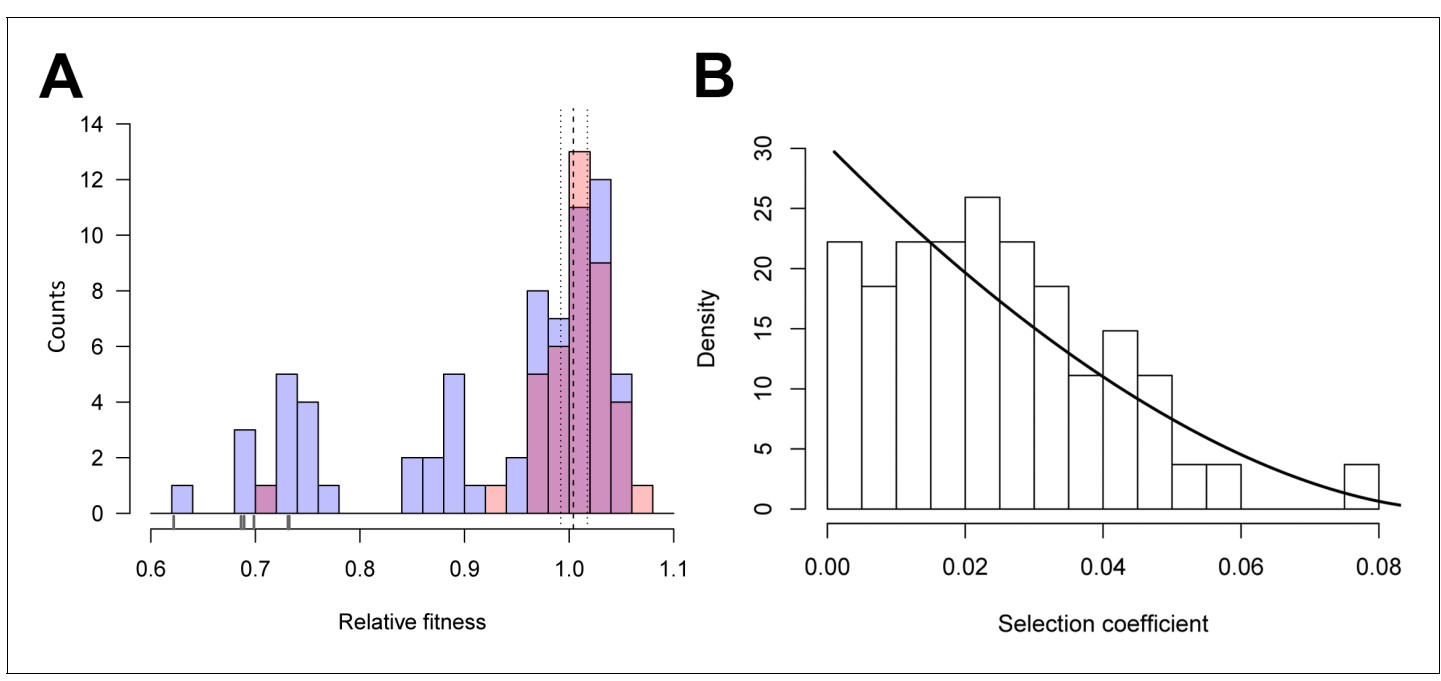

**Figure 1.** Distributions of relative fitness effects of *gtsB* point mutations in low glucose media. (**A**) Counts of nonsynonymous (blue; n = 71) and synonymous (red; n = 39) mutations display a wide range of fitness effects, with ticks under the bars indicating the relative fitness values of nonsense mutations. Dashed and dotted lines show the mean relative fitness of the wild type (WT) competed against the marked competitor. (**B**) The DFE of beneficial-effect mutations (proportions; pooled synonymous and nonsynonymous samples, n = 55) is fit by a κ value of −0.35, which corresponds to the Weibull domain of attraction of the Generalised Pareto Distribution. On this normalised histogram (total area = 1), relative fitness values are shifted to the smallest observed value and expressed as selection coefficients. See *Figure 1—source data 1*.

DOI: https://doi.org/10.7554/eLife.45952.002

The following source data and source codes are available for figure 1:

**Source data 1.** Relative fitness estimates from competitions.
DOI: https://doi.org/10.7554/eLife.45952.003
**Source code 1.** Analysis of distributions of fitness effects.
DOI: https://doi.org/10.7554/eLife.45952.015

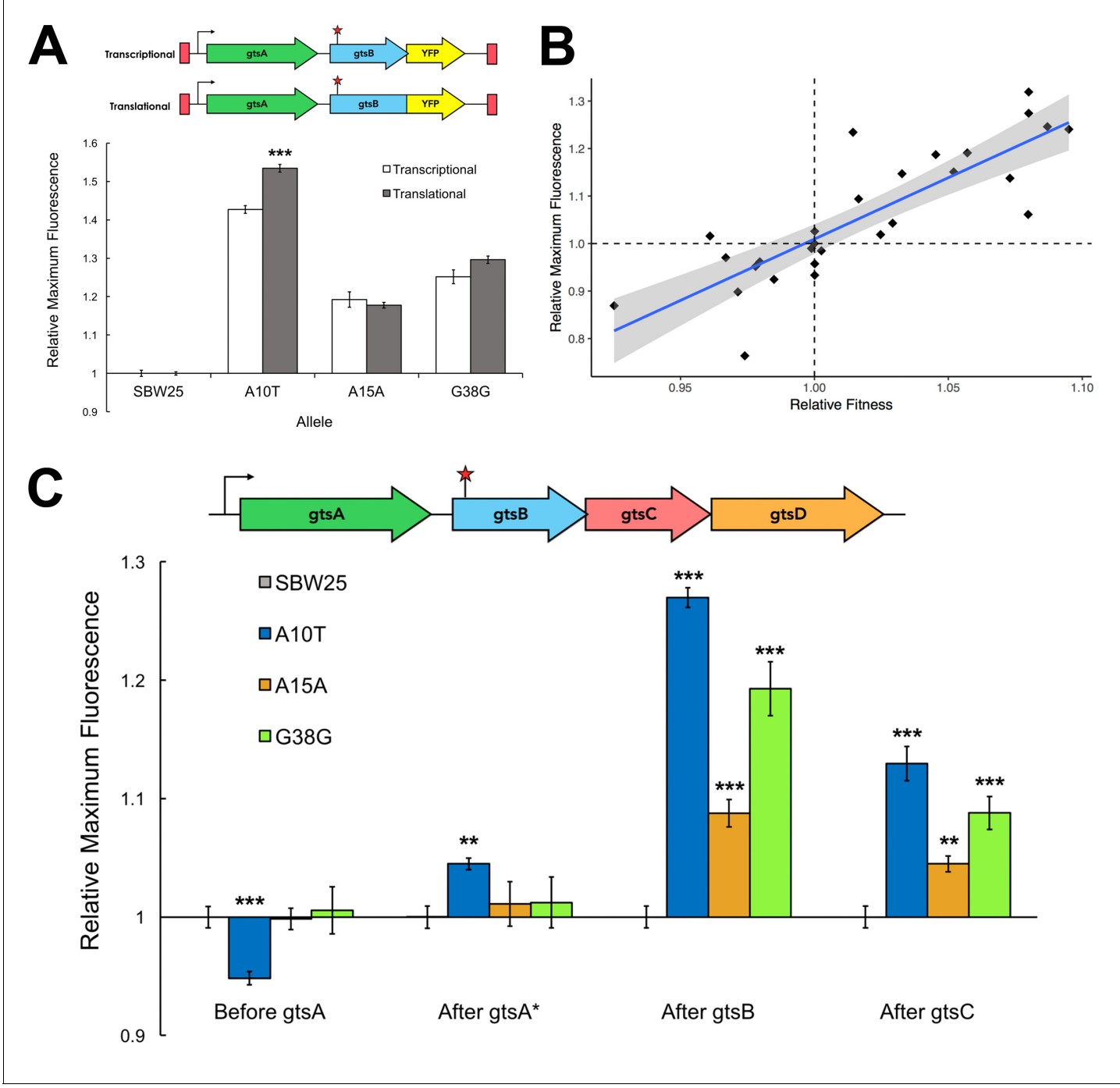

**Figure 2.** Comparison of transcriptional and translational effects of the evolved mutants and correlation with relative fitness. (**A**) The schematic shows the sites of YFP insertion for transcriptional and translational fusions. The plot compares maximum YFP expression (± SEM) from transcriptional and translational YFP fusions at the Tn7 site for the WT (n = 14 replicates) and evolved mutants (n = 7, 7, and six technical replicates, respectively). Significance with respect to transcriptional fusion: ***p<0.001. See **Figure 2—source data 1**. (**B**) Linear regression of fluorescent signal of YFP transcriptional fusions as a proxy for transcript levels and relative fitness measures for a subset of synonymous mutations (n = 27). Grey shading indicates the 95% confidence interval for the regression (adjusted $R^2$ = 0.69, p<0.001). See **Figure 2—source data 2**. (**C**) Expression of transcriptional YFP fusions inserted across the *gts* operon of evolved mutants. Maximum fluorescence (± SEM) of the YFP transcriptional fusions at different loci in the *gts* operon relative to SBW25. See **Figure 2—source data 3**; YFP fusion positions are depicted in **Figure 2—figure supplement 1**. **p<0.01, ***p<0.001.

DOI: https://doi.org/10.7554/eLife.45952.004

The following source data, source code and figure supplements are available for figure 2:

*Figure 2 continued on next page*

*Figure 2 continued*

**Source data 1.** YFP expression for transcriptional and translational fusions after *gtsB* at the Tn7 site.
DOI: https://doi.org/10.7554/eLife.45952.006
**Source data 2.** YFP expression for transcriptional fusions after *gtsB* in the native site for a subset of synonymous mutations.
DOI: https://doi.org/10.7554/eLife.45952.007
**Source data 3.** YFP expression for transcriptional fusions across the *gts* operon.
DOI: https://doi.org/10.7554/eLife.45952.008
**Source code 1.** Source code for *Figure 2C*: analysis of YFP expression and fitness.
DOI: https://doi.org/10.7554/eLife.45952.016
**Figure supplement 1.** Schematic depicting positions of YFP transcriptional fusions within the native *gts* operon.
DOI: https://doi.org/10.7554/eLife.45952.005

with many mutations of small effect and a few of large effect (*Figure 1B*), as expected from theory (*Gillespie, 1984*; *Orr, 2003*; *Martin and Lenormand, 2008*). More formally, the DFE among beneficial mutations is significantly different from an exponential distribution (likelihood ratio test, p=0.0077) and falls within the Weibull domain of the Generalised Pareto Distribution (K = −0.37), suggesting the existence of a local fitness optimum similar to what has been seen previously for non-synonymous mutations (*Rokyta et al., 2008*; *Schoustra et al., 2009*).

What cellular processes underlie the wide range of fitness effects observed here? Our fitness data allow us to test some of the leading hypotheses through in silico analyses. Fitness could be higher if synonymous mutations result in codon usage that is more closely aligned with that of highly expressed genes. Alternatively, it has been suggested that suboptimal codon usage within the first ~50 codons – the translational ramp – is required to ensure efficient translation initiation (*Tuller et al., 2010*; *Navon and Pilpel, 2011*), suggesting that higher fitness should be associated with the introduction of rarer codons close to the start of a gene. We could find no evidence for either explanation in our sample: a regression of fitness on distance from the start codon of *gtsB* was not statistically significant (permutation of residuals, p=0.20), nor was there a significant relationship between change in fitness and change in codon adaptation index (CAI; p=0. 40) or tRNA adaptation index (tAI; p=0.53), both measures of the degree of codon usage bias. This is perhaps unsurprising given that CAI is a gene-level codon usage metric, and a change in a single codon is unlikely to have a large effect on the overall CAI value for either the whole gene or a portion thereof. Notably, there is little evidence for a translational ramp in WT *gtsB*: the first 50 codons are not significantly enriched for rare codons (adj. $R^2$ = 0.0058, p=0.21); further, the interaction of codon position with CAI or tAI does not yield significant results (p=0.45 and 0.90, respectively).

It has also been suggested that synonymous mutations could impact fitness through their effects on mRNA transcript secondary structure and hence the rate and fidelity of translation. Higher fitness could result from faster translation due to transcripts that are less thermodynamically stable and, so, more accessible to the ribosome during translation (*Kudla et al., 2009*) or from more efficient translation due to more stable mRNAs that persist longer due to slower degradation rates (*Deutscher, 2006*). A linear model linking change in mRNA stability and fitness is significant for the nonsynonymous subset of mutations (permutation of residuals, p=0.0039), although the effect is weak (adj. $R^2$ = 0.11) and driven by less stable, highly deleterious mutations. We could not detect a relationship between change in mRNA stability and fitness for synonymous mutations, even when we account for the possibility of strong 5' end secondary structures by adding a position term reflecting distance from the start codon (*Frumkin et al., 2017*).

The absence of any relationship between synonymous mutation fitness and codon usage bias or mRNA stability, both measures affecting translation, suggests that fitness effects stem from changes in transcription. Testing this hypothesis requires comparing estimates of transcript and protein abundance, the difference being a measure of the effect of translation. We evaluated mRNA and GtsB protein levels by proxy via the insertion of a yellow fluorescent protein (YFP) bioreporter into the WT or mutant *gtsB* background just before or just after the stop codon. The former construct (a translational fusion) produces a single reading frame where *gtsB* and YFP are translated together; the latter, with YFP inserted after the *gtsB* stop codon (a transcriptional fusion), results in *gtsB* and YFP being translated separately (*Figure 2A*). A mutation upstream of these fusions that leads to increased translation, but not transcription, is expected to generate a higher YFP expression level in

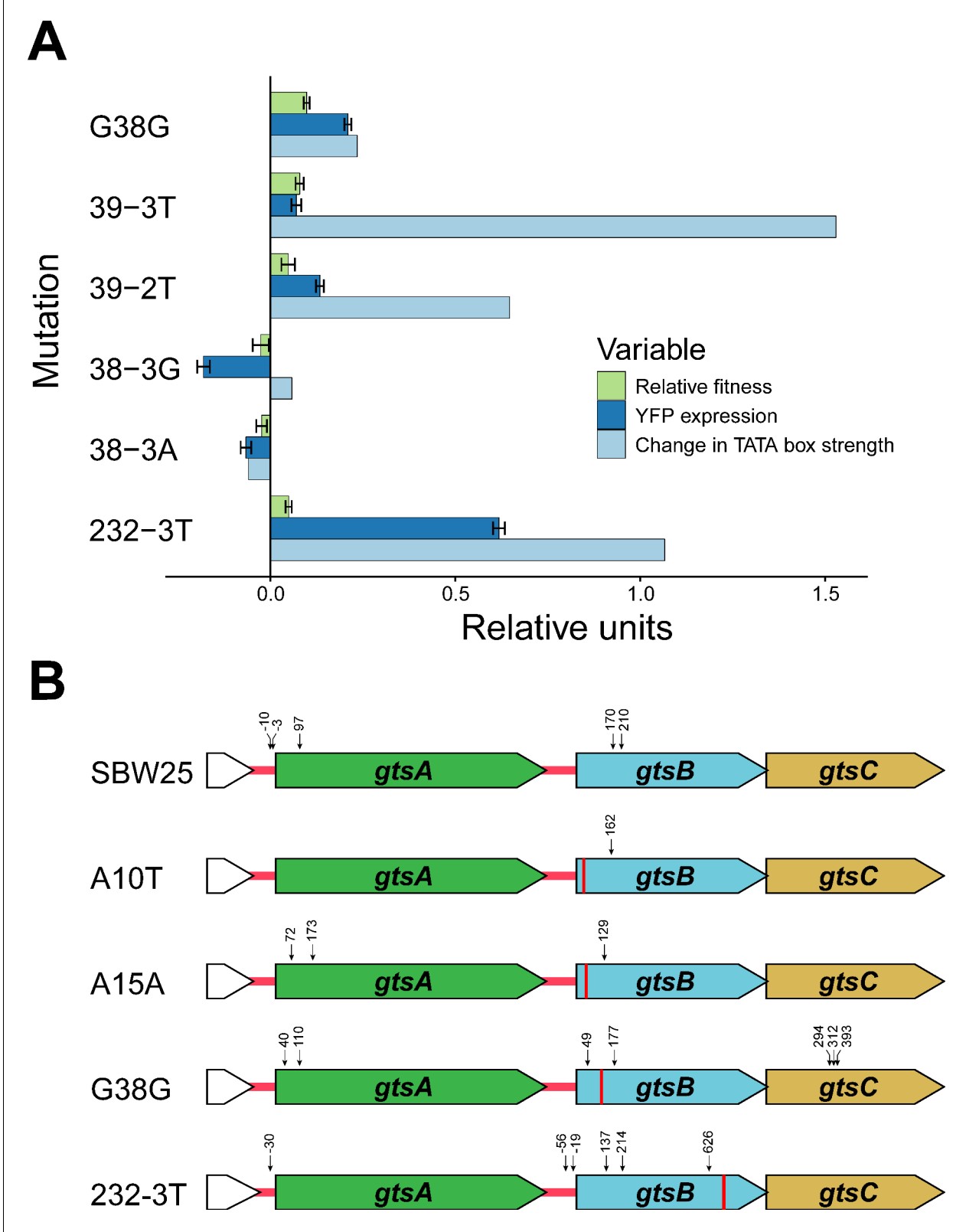

**Figure 3.** Potential mechanism for fitness differences at different loci in *gtsB*. (**A**) Bars represent the mean of each variable in units relative to the WT. Experimental relative fitness and transcriptional expression of YFP measures include standard error. See *Figure 3—source data 1*. (**B**) Locations of potential transcriptional start sites in the *gts* operon are represented by vertical arrows. 5' ends were identified by 5' RACE analysis of RNA isolated from cultures of the wild type (SBW25) and four *gtsB* mutants. The location of each *gtsB* mutation is indicated by a red line, and the nucleotide distance

*Figure 3 continued on next page*

*Figure 3 continued*

from each 5′ end to the nearest start codon is given. See *Figure 3—source data 2*; sequence information for the 5′ RACE analysis is found in *Figure 3—figure supplement 1*.

DOI: https://doi.org/10.7554/eLife.45952.009

The following source data, source code and figure supplements are available for figure 3:

**Source data 1.** YFP, fitness and promoter strength data.

DOI: https://doi.org/10.7554/eLife.45952.011

**Source data 2.** 5RACE experiment results.

DOI: https://doi.org/10.7554/eLife.45952.012

**Source code 1.** Analysis of promoter strength.

DOI: https://doi.org/10.7554/eLife.45952.017

**Figure supplement 1.** Locations of potential transcriptional start sites in the *gts* operon.

DOI: https://doi.org/10.7554/eLife.45952.010

the translational fusion compared to the transcriptional fusion. The expression levels of these different constructs relative to the WT are shown in *Figure 2A* for the two synonymous mutations (A15A, G38G) and a third, independently evolved non-synonymous mutation (A10T) recovered from the original experiment by *Bailey et al. (2014)*. Transcription is elevated in all three mutants relative to the WT but we could not detect any additional effect of translation in the two synonymous mutations, although there is a modest increase in expression associated with translation for the nonsynonymous mutation. These results suggest that these synonymous mutations primarily affect levels of transcription rather than translation.

Two additional lines of evidence point to changes in transcription levels as the likely proximate cause of variation in fitness among our synonymous mutations. First, there is a strong positive relationship between transcript abundance and relative fitness for 27 synonymous mutations (including the A15A and G38G mutations examined above) (*Figure 2B*; $R^2 = 0.691$, p=$1.46 \times 10^{-8}$). Notably, the range of the regression includes mutants with both negative and positive fitness effects, suggesting that the link between transcript abundance and fitness is not limited to beneficial synonymous mutations alone. Second, *Figure 2C* shows that the increased transcription caused by A15A, G38G, and A10T extends downstream to *gtsC* (p<$5.0 \times 10^{-5}$) but not upstream to *gtsA*, which remains largely unaffected (p>0.60). These synonymous mutations thus have polar effects on transcription that extend beyond the gene in which they occur. Taken together with our previous observation that overexpression of WT *gtsB* increases fitness only when the rest of the *gts* operon is also overexpressed (*Bailey et al., 2014*), these results suggest that co-expression of downstream genes is necessary for increased fitness in this system.

What mechanism accounts for the observed changes in transcription and fitness among the synonymous mutations? Previous work has shown that synonymous mutations can generate beneficial effects by creating novel promoters in regions upstream of a gene under selection (*Ando et al., 2014*; *Kershner et al., 2016*). At face value, this mechanism cannot explain our results since we observe a range of fitness effects for synonymous mutations along the entire length of *gtsB*. However, the existence of polar effects on transcription suggests that some synonymous mutations in *gtsB* might be playing a similar role by creating internal promoters causing changes in expression of the downstream genes *gtsC* or *gtsD*. To evaluate this idea, we used Softberry BPROM online software to search the entire *gts* operon for internal sigma 70 bacterial promoter sequences in the ancestral sequence and mutant sequences. We find relatively few hits in our collection, perhaps because BPROM searches for promoters using *Escherichia coli* rather than *P. fluorescens* consensus sequences; however, among the top five hits is a predicted promoter sequence spanning codons 30–42 that includes G38G and 39–3T, the latter being the synonymous mutation with highest fitness in our collection. Both mutations, and an additional beneficial synonymous mutation at 232–3T, result in predicted −10 promoter sequences that are more closely aligned to the −10 consensus sequence for *P. aeruginosa* (TATAAT) than the WT. Notably, there is a tendency for promoter strength to vary positively with both transcription and fitness (*Figure 3A*), although this effect is based on just six mutations and is not significant (permutation of residuals, p=0.17 and 0.11, respectively). These results suggest that fitness changes associated with these synonymous mutations could

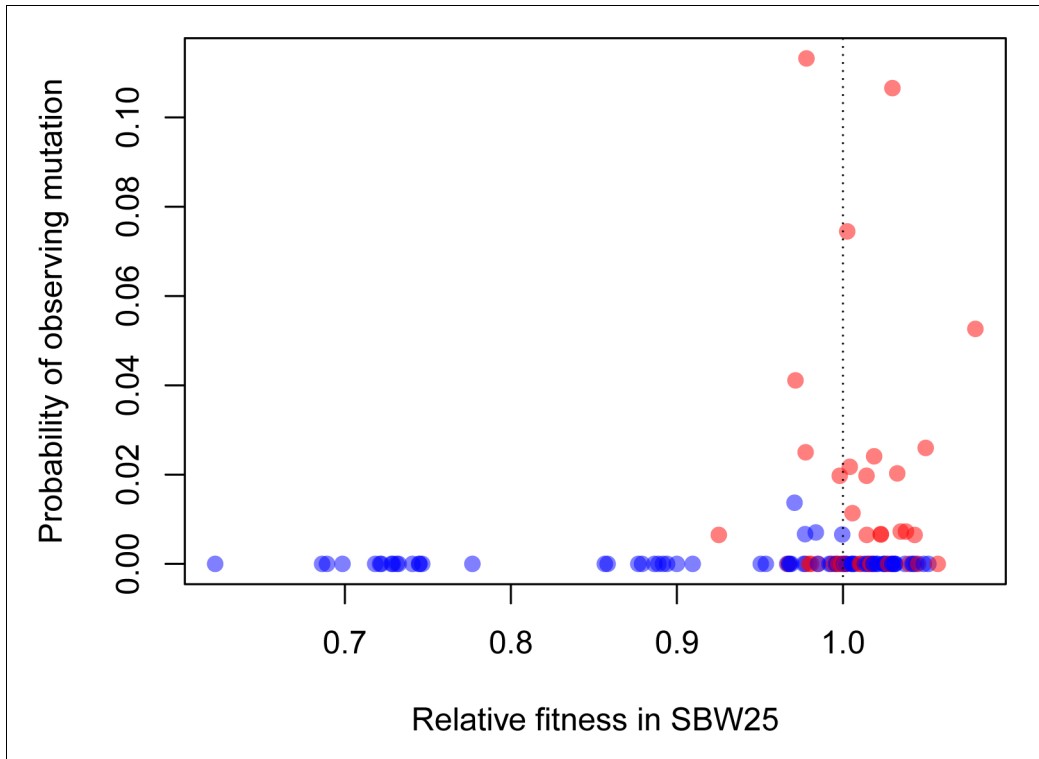

**Figure 4.** Beneficial synonymous mutations (red) are often observed in the phylogeny of related Pseudomonads, while nonsynonymous mutations are less so (blue). There was a significant logarithmic relationship between the probability of observing a given mutation as a binary variable (present/absent) and relative fitness (p=0.0121).
DOI: https://doi.org/10.7554/eLife.45952.013
The following source code is available for figure 4:

**Source code 1.** Phylogenetic analysis.
DOI: https://doi.org/10.7554/eLife.45952.018

be caused by the ability of transcription factors to bind to promoter-like sequences in *gtsB* and alter transcription of downstream genes.

Further support for this interpretation comes from mapping transcriptional start sites for the *gts* operon using a 5' RACE kit for the WT and four additional mutants: G38G and 232–3T (an introduced C → T mutation at the third site of codon 232), the sites identified as being among the top predicted promoters, as well as A10T and A15A which, along with G38G, were recovered from the original experiment in *Bailey et al. (2014)*. The results are summarised in *Figure 3B* and *Figure 3— figure supplement 1*. As expected, we found a transcriptional start site mapping 30 base pairs upstream of the *gtsA* start codon, 25 base pairs downstream of the predicted −10 box. Importantly, and consistent with the hypothesis that mutations in the coding region of *gtsB* can improve promoter binding for the RNA polymerase complex, we mapped transcription start sites 7 and 15 base pairs downstream of the predicted −10 box of the G38G internal promoter, four base pairs downstream of the A15A mutation, and 21 base pairs downstream of the A10T mutation. These results, taken together with an additional transcriptional start site 70 nucleotides upstream of 232–3T, suggest that synonymous mutations in *gtsB* may be strengthening weak internal promoters that were not detected by the available online prediction software. Note that the observation of transcriptional start sites at nucleotide positions near the start of *gtsB* in the WT likely reflects varying degrees of promoter binding strength for the sequences in this region and so is not incompatible with our hypothesis. We also identified transcriptional start sites mapping 19 and 57 base pairs upstream of the *gtsB* start codon within the intergenic space following *gtsA*, implying that *gtsB* may be under independent transcriptional control from *gtsA*.

How often do these synonymous mutations contribute to adaptation in more natural settings beyond the highly contrived conditions we have studied here? We can get part way towards an answer by asking whether fitness in vitro predicts prevalence of a mutation across a phylogeny of pseudomonads. We generated a phylogeny of 77 strains closely related to SBW25 and converted the probability of observing a given mutation to a binary variable based on its presence or absence in the phylogeny while accounting for evolutionary relatedness. For our entire synonymous and non-synonymous sample, we find a positive relationship between the presence of a particular mutation in the phylogeny and its fitness in glucose-limited medium (*Figure 4*, p=0.0210). Notably, our highest fitness mutation, 39–3T, which is synonymous, arises independently multiple times across the phylogeny, even when common ancestry is taken into consideration. These results lend support to the idea that the variation in fitness effects observed here are not an idiosyncratic result of life in a laboratory environment. Rather, the synonymous mutations conferring the highest fitness effects may often contribute to adaptation in more complex, and more natural, environments as well.

## Discussion

Despite mounting evidence to the contrary, the assumption that synonymous mutations are neutral remains deeply embedded in genetics. One need only look as far as the growing number of fitness landscape studies focusing on amino acid replacements which, by definition, involve only nonsynonymous substitutions (*Wu et al., 2016*; *Bank et al., 2016*), the exception being a recent study in yeast showing that most synonymous mutations have small or negligible effects on fitness (*Fragata et al., 2018*). By contrast, our work shows that the DFE among synonymous mutations in *gtsB*, at least, can be highly variable and include both deleterious and beneficial mutations. In fact, aside from the absence of strongly deleterious mutations associated with premature stop codons, the DFE of synonymous mutations in this gene is strikingly similar to that of nonsynonymous mutations and is formally indistinguishable from it if we consider only beneficial mutations. Taken together with the observation of a positive correlation between in vitro estimates of fitness of a given mutation, its prevalence among sequenced isolates, and previous evidence of adaptation via beneficial synonymous mutations (*Bailey et al., 2014*), these results suggest that synonymous mutations in this gene can, and sometimes do, contribute to adaptation.

The cause of the fitness variation among synonymous mutations observed here stems from changes to transcription that impact downstream genes in the same operon. Whether these transcriptional effects occur by changing an internal promoter sequence, as our data suggests, or through some other, still undiscovered mechanisms, remains to be elucidated. It is notable that promoter-associated effects on transcription have been shown to underlie the fitness effects of synonymous mutations in two other microbial systems (*Ando et al., 2014*; *Kershner et al., 2016*), suggesting this mechanism may be quite general, at least for organisms with operon-like genetic architectures. Nevertheless, others have pointed to changes in translational efficiency associated with the accessibility of mRNA near a start codon as the primary mediator of fitness in *Salmonella enterica* (*Kristofich et al., 2018*) and synonymous mutations are known to impact fitness in a wide range of organisms beyond prokaryotes (*Lawrie et al., 2013*; *Cuevas et al., 2012*; *Kashiwagi et al., 2014*). Uncovering the full spectrum of mechanisms by which synonymous mutations impact fitness, and how often they contribute to adaptation, remains a major task for the future.

## Materials and methods

**Key resources table**

| Reagent type (species) or resource | Designation | Source or reference | Identifiers | Additional information |
| --- | --- | --- | --- | --- |
| Strain, strain background (*Pseudomonas fluorescens*) | SBW25; wild type | PMID: 8564013 | | Ancestral strain |

*Continued on next page*

*Continued*

| Reagent type (species) or resource | Designation | Source or reference | Identifiers | Additional information |
|---|---|---|---|---|
| Strain, strain background (*Pseudomonas fluorescens*) | SBW25-*lacZ* | PMID: 17669526 | | SBW25 with neutral chromosomal *lacZ* insertion |
| Strain, strain background (*Escherichia coli*) | DH5α λpir | PMID: 11207743 | | *E. coli* cloning strain |
| Recombinant DNA reagent | pAH79 (plasmid) | PMID: 24912567 | | *P. fluorescens* allelic exchange vector |
| Recombinant DNA reagent | *gtsB* mutagenesis vector (plasmid) | This paper | | pAH79 modified for Golden Gate Assembly of mutant *gtsB* alleles |
| Genetic reagent (*Pseudomonas fluorescens*) | *gtsB* site-directed mutagenesis library | This paper | | Collection of *gtsB* single nucleotide mutants in SBW25 background |
| Recombinant DNA reagent | pUC18T-mini-Tn7T-Gm (plasmid) | PMID: 15908923 | GenBank: AY599232.2 | Source of mini-Tn7T-Gm transposon |
| Recombinant DNA reagent | pUC18T-mini-Tn7T-Gm-eYFP (plasmid) | PMID: 17406227 | GenBank: DQ493879.2 | Source of YFP |
| Genetic reagent (*Pseudomonas fluorescens*) | Mini-Tn7 *gtsB*-YFP transcriptional fusions | This paper | | Transcriptional YFP fusions at SBW25 Tn7 site (see *Figure 2A and B*) |
| Genetic reagent (*Pseudomonas fluorescens*) | Mini-Tn7 *gtsB*-YFP translational fusions | This paper | | Translational YFP fusions at SBW25 Tn7 site (see *Figure 2A*) |
| Genetic reagent (*Pseudomonas fluorescens*) | *gts* operon YFP transcriptional fusions | This paper | | Transcriptional YFP fusions at sites within *gts* operon (see *Figure 2C* and *Figure 2C—figure supplement 1*) |

## Culture conditions

*E. coli* was grown on Luria-Bertani (LB), X-gal sucrose, or tetracycline media. *Pseudomonas fluorescens* SBW25, which was used as the ancestral strain, was grown on LB or X-gal minimal salts media (48 mM $Na_2HPO_4$, 22 mM $KH_2PO_4$, 9 mM NaCl, 19 mM $NH_4Cl$, 2 mM $MgSO_4$, 0.1 mM $CaCl_2$) with glucose (53 µM), succinate (80 µM) or mannitol (53 µM) as indicated. Media were supplemented with 5-bromo-4-chloro-3-indolyl-b-D-galactopyranoside (X-gal) at 40 mg/ml. Antibiotics were used at the following concentrations: 100 µg/ml nitrofurantoin (Nf), 100 µg/ml ampicillin (Ap), 10 µg/ml tetracycline (Tc).

## Molecular cloning

Polymerase chain reactions (PCR) were performed with Phusion High-Fidelity DNA Polymerase (Thermo Fisher Scientific) using custom oligonucleotide primers (Invitrogen) and SBW25 genomic templates (Promega Wizard DNA Extraction Kit). PCR products were purified with the Wizard SV Gel and PCR Cleanup System (Promega). Plasmid DNA was isolated from *E. coli* cultures using the QIAprep Spin Miniprep Kit (Qiagen). Restriction endonucleases and T4 DNA ligase were purchased from New England Biolabs.

Golden Gate assembly reactions (*Engler et al., 2008*) contained approximately equimolar amounts (~20–40 fmol) of destination vector and purified PCR products, 1 µl of 10X T4 ligase buffer, 0.5 µl (200 units) of T4 ligase, and 0.5 µl (10 units) of BsaI enzyme in 10 µl reactions, with incubation for 2 hr at 37°C, 5 min at 50°C, and 5 min at 80°C. Traditional restriction enzyme cloning was performed according to standard protocols, with separate digestion of vector and insert DNA (2 hr at 37°C) followed by spin column purification and overnight ligation at 16°C. Ligation reactions were transformed into chemically-competent *E. coli* DH5α λpir by the Inoue method (*Sambrook et al., 1989*).

## Construction of *gtsB* mutagenesis vector

The *P. fluorescens* allelic exchange vector pAH79 (*Bailey et al., 2014*) was modified for rapid generation of mutant *gtsB* alleles by Golden Gate assembly (GGA) of polymerase chain reaction (PCR) amplicons. A three-part ligation between a digested pAH79 derivative (BglII and SpeI), *lacZα* amplicon (BglII and MfeI), and SBW25 amplicon spanning 114 to 865 bp downstream of the *gtsB* stop codon (MfeI and XbaI) yielded the final *gtsB* mutagenesis vector. The vector includes two BsaI cloning sites compatible for Golden Gate assembly of PCR products amplified with primers F2-gtsB-F and R3-gtsB-R (*Table 1*) in combination with mutagenic primers (Tables S1 and S2 in *Supplementary file 1*).

## Site-directed mutagenesis of *gtsB*

Site-directed mutagenesis of *gtsB* (*PFLU4845*) was accomplished by cloning *gtsB* alleles with a single mutation into the mutagenesis vector described above, generating an *E. coli* library, followed by allelic replacements in SBW25. Primers (*Supplementary file 1*) were designed to introduce mutations at 112 sites spanning the *gtsB* gene, at every tenth codon along the gene and saturating the sites neighbouring three previously identified beneficial mutations. Sequences from 715 base pairs upstream to 173 base pairs downstream of *gtsB* were amplified as two PCR fragments, one of which contained a threefold degenerate polymorphism introduced by a mutagenic primer. BsaI recognition sequences were included in each primer to enable seamless ligation between the PCR products and mutagenesis vectors using GGA (*Engler et al., 2008*). Cloning reactions were transformed into *E. coli* DH5α λpir with selection on ampicillin.

Transformations yielded libraries of *E. coli* strains for introduction of mutations into SBW25. Recombination of each mutant *gtsB* allele into the chromosome was selected for in two steps: selection for Tc$^R$ followed by selection for sucrose resistance as previously described (*Bailey et al., 2014*). We used an SBW25 recipient strain in which the native *gtsB* was replaced by *lacZ*, allowing us to use blue-white screening on LB 5% sucrose X-gal agar to identity recombinants in which *lacZ* was replaced by the vector-encoded mutant *gtsB*. The sucrose-resistant white colonies were used as PCR templates for amplification of the *gtsB* locus using an M13F-tagged primer (*Table 1*), for sequencing by the McGill University and Genome Quebec Innovation Centre.

**Table 1.** Oligonucleotides used in this study.

Restriction enzyme recognition sequences are capitalised. BsaI overhangs are underlined. Introduced mutations are in bolded capital letters. Additional oligonucleotides used for site-directed mutagenesis are listed in Tables S1 and S2 of *Supplementary file 1*.

| Name | Sequence (5' to 3') | Function |
|---|---|---|
| F2-pUC19-BsaI | gcgAGATCTgtcgtGAGACCggtgatgacggtgaaaacct | *gtsB* mutagenesis vector construction |
| R3-pUC19-MfeI-SpeI | actgcgACTAGTCAATTGattaatgcagctggcacgac | *gtsB* mutagenesis vector construction |
| F-800Right | actgcgCAATTGagaccccggaagacatcag | *gtsB* mutagenesis vector construction |
| R-800Right | actgcgTCTAGAcattgcgaagttcaagcgta | *gtsB* mutagenesis vector construction |
| F2-gtsB-F | actgcgGGTCTCagtcgaaaagtcgcgacctacatgg | Conserved *gtsB* forward primer |
| R3-gtsB-R | actgcgGGTCTCctgccggaCaccacggtcggccagctc | Conserved *gtsB* reverse primer |
| 4845-M13F | GTAAAACGACGGCCAGTTCCGACAGGCTGTAGTCCTT | *gtsB* sequencing primer |
| R2-M13R-gtsB | GGAAACAGCTATGACCATGTGGTCCTCAGCTCGGAATA | *gtsB* sequencing primer |
| SP1 | ACCACACCGAACAGGAAGTC | 5' RACE cDNA synthesis |
| B-SP2 | ACTGCGTCTAGAGACCAAGGTGATACCGATAAACA | 5' RACE *gtsB* amplification |
| B-SP3 | ACTGCGTCTAGACGAACAAGGCCAGGTTTTT | 5' RACE *gtsB* amplification |
| A-SP2 | ACTGCGTCTAGATTTCTTGTCGAGCAGGGAGT | 5' RACE *gtsA* amplification |
| A-SP3 | ACTGCGTCTAGATTCTTCTTTGGCGACGTCTT | 5' RACE *gtsA* amplification |

DOI: https://doi.org/10.7554/eLife.45952.014

## DNA extraction and whole genome sequencing

Strains were grown in LB liquid media overnight; genomic DNA was extracted using the QIAGEN DNeasy Blood and Tissue Kit. Sequence data were generated on the Illumina MiSeq platform with paired-end reads using the Nextera XT kit. Reads generated were approximately 300 bp in length.

## Reference-based mapping and variant calling

The genome of *P. fluorescens* SBW25 is publicly available from NCBI (PRJEA31229). A modified version of the bioinformatics pipeline described in *Dettman et al. (2012)* was used to analyse the reads. Briefly, reads were trimmed using Popoolation (ver. 1.2.2; *Kofler et al., 2011*) with a Phred quality threshold of 20 and a minimum retention length of 75% of original read length. Trimmed reads were then mapped to the SBW25 reference genome using Novoalign (ver. 3.02.08, www.novo-craft.com). Single nucleotide polymorphisms and indels were annotated using Samtools (ver. 1.9; *Li et al., 2009*), BCFtools (ver. 1.9), VarScan (ver. 2.3.7; *Koboldt et al., 2012*), and snpEff (ver. 4.0; *Cingolani et al., 2012*). Read and alignment quality were assessed using FastQC (ver. 0.11.7 www.bioinformatics.babraham.ac.uk/projects/fastqc/). Sequence data are available from the NCBI Short Read Archive under BioProject PRJNA515918: Pseudomonas fluorescens SBW25 gtsB mutants.

## Competitions

Competitions were performed as outlined in *Lenski et al. (1991)* on four to six replicates (genetically identical clones) of 110 mutant strains. This method encompasses all growth phases of bacterial culture, including lag and exponential growth. These replicates provide a measure of the variability inherent in our experimental procedure, and are thus considered technical replicates. All strains, including SBW25-*lacZ*, were removed from storage at −80℃ and grown overnight at 28℃ on LB agar. Single colonies were inoculated into 2 mL LB broth for overnight incubation at 28℃ under shaking conditions. Each mutant strain was transferred into minimal glucose media for a 24 hr acclimation period at 28℃, then mixed in a 1:1 volumetric ratio with SBW25-*lacZ* and inoculated into 2 mL of minimal media with glucose. Initial and final aliquots from mixed cultures were frozen in 20% glycerol after 1 and 24 hr' growth and plated on minimal agar with glucose. Only plates containing 30 or more colonies of each strain were included. Relative fitness was calculated using $w = (f_{final}/f_{initial})(1/doublings)$, where $f_{final}$ and $f_{initial}$ are ratios of the frequency of mutant to the frequency of SBW25-*lacZ* strain after and before the competition. The number of doublings was estimated from the dilution factor and corresponded to ~6.7 or 13.2 generations depending on the dilution factor. The effect of the *lacZ* marker was tested by competing SBW25-*lacZ* against the WT with each batch of competitions. The mean relative fitness of SBW25-*lacZ* was $1.005 \pm 0.0007$ SEM.

## Estimating codon preference and mRNA stability

In order to estimate the change in codon bias attributable to each synonymous mutation, we compared the CAI value of the mutant to the WT using SBW25 ribosomal protein genes as a reference (*Sharp and Li, 1987*). The 'cai' function in the 'seqinr' package in R (*Charif and Lobry, 2007*) was used to calculate change in CAI at each site. tAI values were calculated by inputting tRNA gene copy number, a proxy for tRNA expression (*Tuller et al., 2010*), in the stAI$_{calc}$ interface (*Sabi et al., 2017*). As per previous work (*Kudla et al., 2009*), we predicted the most likely folding energy of 42-nucleotide windows centred on each mutation using the 'mfold' server (*Zuker, 2003*).

## Comparison and characterisation of DFEs

All statistical analyses were conducted in R Studio (version 1.0.136; www.rstudio.com). Six nonsense mutations were omitted from our analysis, since they likely do not result in a complete protein. We compared synonymous and nonsynonymous DFEs for all mutations and for the subset of beneficial mutations by bootstrapping the Kolmogorov-Smirnov (K-S) statistic. We found no significant difference between K-S values for nonsynonymous and synonymous beneficial mutations, so we pooled the data to infer the properties of the tail distribution following the method outlined in *Beisel et al. (2007)*. Relative fitness values were transformed to selection coefficients by subtracting 1; we shifted the threshold to the smallest observed selection coefficient, as suggested by *Beisel et al. (2007)*. Using the 'GenSA' package in R, we estimated the optimal value of the scale parameter τ, which characterises the stretch of the distribution, with κ (the tail parameter) set to 0, corresponding to an

exponential distribution; in the alternative model, optimal τ and κ values were calculated without restricting κ. A likelihood ratio test was used to determine whether the model with the unconstrained κ value was a better fit than the exponential distribution.

### Statistical analysis

Permutation of residuals was used as per *Still and White (1981)* to test for significant relationships between explanatory variables and fitness. We tested for significant differences in YFP expression between mutant and WT alleles using a two-tailed T-test assuming equal variance. Threshold for significance was $\alpha = 0.05$.

### Transcriptional and translational fusion of YFP at Tn7 site

Transcriptional and translational fusion constructs were generated using GGA (*Engler et al., 2008*) and the use of site-specific mini-Tn7 transposon and yellow fluorescent protein (YFP) sequences (*Choi et al., 2005*; *Choi and Schweizer, 2006*). A 2.6 kb PCR product was amplified from genomic template DNA containing the target locus, and included the 346 bp promoter region of *gtsA*, the open reading frame of *gtsA* and the open reading frame of *gtsB*. This PCR product and the downstream YFP fusion PCR product were seamlessly ligated together into a derivative of the Tn7 vector pUC18T-mini-Tn7T-Gm through GGA.

The YFP transcriptional fusion after *gtsB* required additional modifications due to the eight nucleotide overlap between the stop codon of *gtsB* and start codon of *gtsC*. To preserve the *gtsB* sequence and predicted *gtsC* ribosomal binding site, the start codon of the YFP transcriptional fusion was inserted in-frame after the first four codons *gtsC*. Translational YFP fusions had a 6-glycine ((GGC)6) linker sequence (*Chen et al., 2013*) between the second-last codon (302) of *gtsB* and the second codon of YFP, which removed both the stop codon of *gtsB* and start codon of YFP to create a single peptide. The 3' UTR of both the transcriptional and translational YFP fusions includes an intrinsic transcriptional terminator. Tn7 vectors were transformed into *E. coli* DH5α λpir following the Inoue method, then incorporated into SBW25 and verified as described in *Bailey et al. (2014)*.

### YFP transcriptional fusions at the native *gts* operon

YFP transcriptional fusions across the *gts* operon were constructed using an allelic replacement strategy similar to the site-directed mutagenesis method described above. YFP sequences were PCR amplified from a YFP storage vector, and sequences upstream and downstream of the desired YFP insertion locus were amplified from SBW25 template DNA as two PCR products ranging from 400 to 900 base pairs in length. Upstream PCR products included predicted native ribosomal binding sites to enable YFP translation. Forward and reverse primers for all PCR products were engineered with BsaI recognition sites to allow for 4-part seamless ligation by GGA (*Engler et al., 2008*) of the YFP PCR product and upstream and downstream flanking PCR products to an GGA allelic replacement vector derived from pAH79 (*Bailey et al., 2014*). YFP fusions were transformed into *E. coli* DH5α λpir and recombined into *P. fluorescens* SBW25 by two-step allelic replacement as previously described (*Bailey et al., 2014*). Fusion junctions and *gtsB* mutations were confirmed through diagnostic PCR and Sanger sequencing.

### Glucose induction assays

Cultures were inoculated from individual colonies (biological replicate) and grown shaking at 28°C in 200 μL minimal (M9) media supplemented with 25.6 mM of succinate as the sole carbon source. After 24 hr of growth, 20 μL of culture was transferred to 180 μL of minimal (M9) media supplemented with 212 μM glucose in a transparent 96 well plate. The 96 well plate was incubated for 10 hr static at 28°C in a Tecan Infinite M200 Pro fluorescent plate reader where optical density (OD) and fluorescence measurements were taken every 10 min. OD was measure by absorbance at 595 nm wavelength and fluorescence was measured with 500 nm excitation and 535 nm emission wavelengths. The maximum fluorescence was calculated as the maximum YFP signal (~7 hr), subtracted by the background fluorescence of unmarked SBW25 and then standardised by dividing by the blank corrected OD. Maximum fluorescence values were then divided by the ancestral SBW25 values to determine relative effect sizes.

## RNA isolation and identification of transcriptional start sites

Isolated colonies were cultured in 1.7 mL of M9 media supplemented with 25.6 mM succinate and incubated overnight at 28˚C shaking. Vials with 7.2 mL of M9 media supplemented with 212 µM glucose were inoculated with 800 µL of culture and incubated at 28˚C shaking. $OD_{600}$ was measured every 15 min and cells were harvested at mid-log phase (2 hr) by centrifugation at 6770 x $g$ for 10 min at 4˚C. Pellets were resuspended in 500 µL M9 media and treated with RNAprotect Bacteria Reagent (Qiagen). RNA was isolated using RNeasy Mini Kit (Qiagen) with additional DNase treatment with RNase-free DNase Set (Qiagen). RNA concentration was quantified using NanoDrop 2000 Spectrophotometer (Thermo Scientific). cDNA synthesis, cDNA purification and poly(A) tailing were completed using 5'/3' Random Amplification of cDNA ends (RACE) kit, 2nd generation (Roche) and Wizard SV Gel and PCR Clean-Up System (Promega). First strand $gts$ operon cDNA was generated using primer SP1 and cDNA was amplified using Phusion High-Fidelity DNA Polymerase (Thermo Fisher Scientific). First and second rounds of amplification of dA-tailed cDNA used primers B-SP2 and B-SP3 for $gtsB$ an A-SP2 and A-SP3 for $gtsA$ respectively (see *Table 1* for primer sequences). 2nd round amplification products were run on 2% agarose gel and expected bands (150–400 base pairs) were excised and purified using Qiaquick Gel Extraction Kit (Qiagen). DNA fragments were digested with SalI-HF and XbaI and cloned into a pUC19 vector using T4 DNA Ligase (NEB). Inserted DNA fragments were sequenced using Sanger sequencing to identify possible transcriptional start sites in $gtsA$ and $gtsB$. Sequences were aligned to the $gts$ operon using CodonCode Aligner.

## Phylogeny construction

A phylogeny was constructed using full DNA sequences of $rpoB$, $rpoD$, and $gyrB$ for 77 closely related Pseudomonas strains obtained from NCBI, as per *Gomila et al. (2015)*. The concatenated sequences were aligned using NCBI's BLAST. MEGA7 (*Kumar et al., 2016*) was used to build the tree based on the aligned concatenated sequences using maximum likelihood to generate a bootstrapped consensus tree ($n$ = 500). For each site in the $gtsB$ alignment, maximum parsimony was used to estimate the ancestral state at all internal nodes in the phylogeny with the function 'ancestral.pars' from the phangorn package in R (*Schliep, 2011*). As our phylogeny had polytomies, the ancestral states were estimated for 100 randomly resolved bifurcating phylogenies and the frequency of the inferred ancestral state at each internal node was calculated. For each site in $gtsB$, the number of evolutionary events was calculated by comparing the inferred state at the beginning and end of each branch and counting the number of transitions. A binary model was fit expressing whether mutations of interest were observed in the phylogeny.

## Acknowledgements

Thanks to N Rodrigue, A Schick, and A Wong for feedback and discussion. Funding: This work was supported by a Natural Sciences and Engineering Research Council (Canada) Discovery Grant to RK, an NSERC Canada Graduate Scholarship to ELT, and an Ontario Graduate Scholarship to NM. Competing interests: Authors declare no competing interests. Data and materials availability: Source data files for all figures are linked to this publication. Genomic data has been deposited into the NCBI Sequence Read Archive as BioProject PRJNA515918.

## Additional information

### Funding

| Funder | Grant reference number | Author |
| --- | --- | --- |
| Natural Sciences and Engineering Research Council of Canada | Discovery Grant | Rees Kassen |
| Natural Sciences and Engineering Research Council of Canada | Canada Graduate Scholarship | Eleonore Lebeuf-Taylor |
| Ontario Ministry of Economic Development and Innovation | Ontario Graduate Scholarship | Nick McCloskey |

The funders had no role in study design, data collection and interpretation, or the decision to submit the work for publication.

## Author contributions
Eleonore Lebeuf-Taylor, Data curation, Formal analysis, Validation, Investigation, Visualization, Methodology, Writing—original draft, Writing—review and editing; Nick McCloskey, Data curation, Validation, Investigation, Visualization, Methodology, Writing—original draft; Susan F Bailey, Conceptualization, Data curation, Formal analysis, Validation, Investigation, Visualization, Methodology, Writing—review and editing; Aaron Hinz, Conceptualization, Validation, Investigation, Methodology, Writing—original draft; Rees Kassen, Conceptualization, Resources, Formal analysis, Supervision, Funding acquisition, Investigation, Visualization, Methodology, Writing—original draft, Project administration, Writing—review and editing

## Author ORCIDs
Eleonore Lebeuf-Taylor (iD) https://orcid.org/0000-0001-9082-6049
Susan F Bailey (iD) https://orcid.org/0000-0002-2294-1229
Rees Kassen (iD) https://orcid.org/0000-0002-5617-4259

## Decision letter and Author response
Decision letter https://doi.org/10.7554/eLife.45952.024
Author response https://doi.org/10.7554/eLife.45952.025

# Additional files

## Supplementary files
• Supplementary file 1. Sequences of *gtsB* mutagenesis primers are found in Tables S1 and S2.
DOI: https://doi.org/10.7554/eLife.45952.019
• Transparent reporting form
DOI: https://doi.org/10.7554/eLife.45952.020

## Data availability
Genomic data has been deposited into the NCBI Sequence Read Archive as BioProject PRJNA515918. All other data generated during this study are included in the manuscript and supporting files. Source data files have been provided for Figures 1, 2, and 3.

The following dataset was generated:

| Author(s) | Year | Dataset title | Dataset URL | Database and Identifier |
|-----------|------|---------------|-------------|-------------------------|
| Lebeuf-Taylor E, McCloskey N, Bailey SF, Hinz A, Kassen R | 2019 | Pseudomonas fluorescens SBW25 gtsB mutants | https://www.ncbi.nlm.nih.gov/bioproject/PRJNA515918 | NCBI Sequence Read Archive, BioProject PRJNA515918 |

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
