## [Decision Letter]

Thank you for submitting your article "The distribution of fitness effects among synonymous mutations in a gene under selection" for consideration by *eLife*. Your article has been reviewed by two peer reviewers, one of whom is a member of our Board of Reviewing Editors, and the evaluation has been overseen by Patricia Wittkopp as the Senior Editor. The following individuals involved in review of your submission have agreed to reveal their identity: Sandeep Venkataram (Reviewer #2).

The reviewers have discussed the reviews with one another and the Reviewing Editor has drafted this decision to help you prepare a revised submission.

This is a very interesting paper and a potentially important contribution to the field of evolution. The study provides a significant contribution to our knowledge of the fitness effects of synonymous variants and uncovers possible mechanisms of action of adaptive synonymous mutations. The reviewer's comments are summarized here.

1) As the authors state, the fitness changes associated with synonymous mutations could be caused by TF binding capacity. The sign of the changes on Figure 3 are correlated but not so much the magnitude. It would have been useful to also have additional experiments or analyses strengthening the results, for instance the introduction of other mutations that would be predicted from promoter sequences to increase and decrease binding and see how it affects fitness. An examination of the data suggests that most of the correlation between TATA box strength and YFP expression seems to be driven by 39-3T and 38-3G, which have the largest mismatches between expression and TATA box strength. Also, it is not clear if all the mutations considered are synonymous since 39-2T maybe a non-synonymous one (Tyr -> Phe, correct?). Therefore, the causal link between synonymous mutations, expression levels and fitness remains to be completely shown and most importantly the link with the creation of a novel promoter element. This is critical for the study as the mechanism linking synonymous mutations to fitness is one of the most striking findings of the paper.

2) Another important aspect could be to try to see how frequent such synonymous mutations with strong impact could be across the genome, i.e. how easy it would be in general to create transcription factor binding sites in coding sequences. This could help generalize the results.

3) The Abstract states: "We used site-directed mutagenesis coupled with direct measures of competitive fitness to estimate the distribution of fitness effects among synonymous mutations for a gene under selection." However, the authors have prior information that the gene they have chosen to characterize can adapt via synonymous mutations from their prior work, making it unclear if the measured distribution is truly representative of "the" synonymous DFE. This also carries over to their primary conclusion that the "DFE of synonymous mutations is strikingly similar to that of nonsynonymous mutations and is formally indistinguishable from it if we consider only beneficial mutations". While this does not detract substantially from the impact of the work, as the identification of a large number of adaptive synonymous mutations in any gene has a significant impact on the typical assumption that synonymous variants are selectively neutral, these sentences should be reworded to reflect this limitation. If the gene chosen for this study is an exceptional case, the DFE established remains true but regards a very small number of genes. At minimum, the wording should be changed throughout the paper and this aspect discussed in the Discussion section.

4) Another concern related to 3) about the manuscript is that some statements are too strong, for instance in the first paragraph of the Discussion it says that DFE among synonymous mutations can be highly variable contrary to what has often been assumed. Is it still the case in the community? It would be important to substantiate this with references showing that it is still the case.

5) Regarding the use of the codon adaptation index (CAI) as a metric of adaptation for codon usage. CAI is that it is a metric that looks for gene-level codon usage differences by comparing the distribution of codons used within the gene to a reference set. As the gene used in the study consists of ~300 codons, changing a single codon through a point mutation is highly unlikely to ever result in a significant change in CAI as it has little impact on the total codon distribution of the gene. The lack of association is therefore not surprising. As an alternative method for testing for codon usage adaptation, the authors could quantify the difference in ancestral and mutant codon usage for the specific mutated codon and compare that difference to the fitness benefit shown by that isolate. If adaptation is driven by codon usage, one would expect a significant positive correlation between the gain in codon preference at the mutated site and fitness. The use of this or any similar metric that specifically focuses on the mutated codons would substantially improve the work. Note that these additional experiments would strengthen the paper but are not required. Nevertheless, the limitation of the CAI metric used should be taken into account.

6) One piece of information that may not have been taken full advantage of is the difference between the number of sites targeted for mutagenesis (112) and the number recovered (34). Assuming that the authors sampled a large number of clones from the transformed library (say 1000+), the loss of sites from the sampled mutants means that mutation at these sites is lethal. This could add significant new information into the deleterious region of the DFE. If a saturating number of clones was not sampled, it should be made clear in the Materials and methods.

7) The phylogenetic analyses showing that the 38-3T has arisen independently should be detailed because the correlation shown relies on the fact that these mutations are independent. If the points shown on Figure 4 are not independent, i.e. some strains share the mutations because of common ancestry and not because it was acquired, the correlation could derive from other factors also correlated with this ancestry.

---

## [Author Response]

This is a very interesting paper and a potentially important contribution to the field of evolution. The study provides a significant contribution to our knowledge of the fitness effects of synonymous variants and uncovers possible mechanisms of action of adaptive synonymous mutations. The reviewer's comments are summarized here.1) As the authors state, the fitness changes associated with synonymous mutations could be caused by TF binding capacity. The sign of the changes on Figure 3 are correlated but not so much the magnitude. It would have been useful to also have additional experiments or analyses strengthening the results, for instance the introduction of other mutations that would be predicted from promoter sequences to increase and decrease binding and see how it affects fitness. An examination of the data suggests that most of the correlation between TATA box strength and YFP expression seems to be driven by 39-3T and 38-3G, which have the largest mismatches between expression and TATA box strength. Also, it is not clear if all the mutations considered are synonymous since 39-2T maybe a non-synonymous one (Tyr -> Phe, correct?). Therefore, the causal link between synonymous mutations, expression levels and fitness remains to be completely shown and most importantly the link with the creation of a novel promoter element. This is critical for the study as the mechanism linking synonymous mutations to fitness is one of the most striking findings of the paper.

We agree with the reviewers that the results of the in silico analysis are consistent with the novel promoter mechanism but perhaps not as compelling as we would like. We therefore considered a range of options to obtain a more direct test of this hypothesis (which, by the way, is not restricted to synonymous mutations because nonsynonymous mutations can also change promoter binding sequences) including 5’RACE, RNA seq and genetic approaches. We settled on 5’RACE because it is a well-established tool for identifying transcription start sites and so would allow us to assess directly the link between *gtsB* mutations and transcription initiation. An RNA seq experiment was not feasible within the timeframe and resources available to us, not only because of the additional effort involved in carrying out the experiment and obtaining sequencing but also because we would need to spend substantial time on optimization, as the low glucose concentrations in our experiment demand we pool replicate cultures to obtain sufficient RNA for sequencing (this was also the reason that we reverted to the YFP-fusion approach to estimate transcriptional and translational effects of our mutations), and introducing an additional source of error.

The 5’RACE work focuses on the WT and four mutant genotypes, three with synonymous and one with a nonsynonymous mutation. Consistent with our hypothesis, we find transcriptional start sites just downstream of A10T, A15A, and G38G and just upstream of 232-3T. The WT shows transcriptional start sites towards the start of the gene as well (though not at the same nt positions as any of the mutants), a result that likely reflects the fact that the sequences in this region have varying degrees of promoter binding strength. 5’RACE is good at identifying transcription initiation sites but does not provide any measure of promoter binding strength. The observation of transcript initiation in both the WT and the mutants is therefore not evidence against our hypothesis. Interestingly, this analysis revealed additional transcriptional start sites in the intergenic region between *gtsA* and *gtsB*, suggesting *gtsB* may be under separate transcriptional control from *gtsA*. Moreover, the spectrum of transcriptional start sites along the entire length of *gtsB* associated with the 232-3T mutation underscores the limitations of the *in silico* analysis; there is clearly much that is being missed with this approach (see response to point 2 below).

Taken together, the results of the 5’RACE experiments provide additional, experimental support for our hypothesis. We acknowledge that a more comprehensive test will require further work, including both RNA seq and genetic approaches, to unpack in more detail the architecture of the operon. This extra work is beyond the scope of the present manuscript, however, given that our work provides insights extending across a wide range of biological organization, from the molecular causes of fitness variation among synonymous mutations though to the prevalence of these mutations among natural isolates.

2) Another important aspect could be to try to see how frequent such synonymous mutations with strong impact could be across the genome, i.e. how easy it would be in general to create transcription factor binding sites in coding sequences. This could help generalize the results.

This is an interesting suggestion but, we feel, beyond the scope of this paper for two reasons. The first is that, as our 5’RACE results underscore, the promoter sequences identified by current in silico approaches are not necessarily those that *P. fluorescens* uses. Without a better understanding of what the consensus sequence is for this strain, we would have little confidence in our results. The second, and related to the first, is that identifying these sequences and doing the analysis is a project unto itself. An alternative approach, which we are pursuing, is to extend the phylogenetic analysis reported in our paper to entire genomes. This requires very large sample sizes (hundreds of strains at least) and significant computational resources. Please stay tuned for the results in the next year or so.

3) The Abstract states: "We used site-directed mutagenesis coupled with direct measures of competitive fitness to estimate the distribution of fitness effects among synonymous mutations for a gene under selection." However, the authors have prior information that the gene they have chosen to characterize can adapt via synonymous mutations from their prior work, making it unclear if the measured distribution is truly representative of "the" synonymous DFE. This also carries over to their primary conclusion that the "DFE of synonymous mutations is strikingly similar to that of nonsynonymous mutations and is formally indistinguishable from it if we consider only beneficial mutations". While this does not detract substantially from the impact of the work, as the identification of a large number of adaptive synonymous mutations in any gene has a significant impact on the typical assumption that synonymous variants are selectively neutral, these sentences should be reworded to reflect this limitation. If the gene chosen for this study is an exceptional case, the DFE established remains true but regards a very small number of genes. At minimum, the wording should be changed throughout the paper and this aspect discussed in the Discussion section.

Thank you for flagging this for us. We certainly did not intend to give the impression that our work represents ‘the’ DFE for all synonymous mutations. Rather, as the reviewers point out, we focus on the DFE among synonymous mutations in this one gene in particular, where we have prior evidence that adaptation can occur through synonymous mutations. We have revised the manuscript to make it clearer that the DFE we are estimating is for this gene only. Specifically, we have made the following changes:

The sentence in the Abstract now reads, “We used site-directed mutagenesis coupled with direct measures of competitive fitness to estimate the distribution of fitness effects among synonymous mutations in a gene under directional selection and capable of adapting via synonymous nucleotide changes.”

We have modified our main concluding sentence to read, “We have shown that, contrary to what has often been assumed, the DFE among synonymous mutations, in *gtsB* at least, can be highly variable and include both deleterious and beneficial mutations.”

The concluding sentence of the same paragraph now reads, “Taken together with the observation of a positive correlation between in vitro estimates of fitness of a given mutation, its prevalence among sequenced isolates, and previous evidence of adaptation via beneficial synonymous mutations, these results suggest that synonymous mutations in this gene can, and sometimes do, contribute to adaptation.”

Finally, there is little more we can say about the generality of this result beyond what we have already said in the closing paragraph. We therefore do not feel we can add anything further to this discussion at this time. Broader statements will require more studies examining the DFE among synonymous mutations in a range of genes, genetic architectures, and organisms. We hope our paper will spur others to take on these kinds of studies and confront our long-standing assumptions about the genetic code with more data.

4) Another concern related to 3) about the manuscript is that some statements are too strong, for instance in the first paragraph of the Discussion it says that DFE among synonymous mutations can be highly variable contrary to what has often been assumed. Is it still the case in the community? It would be important to substantiate this with references showing that it is still the case.

The assumption that synonymous mutations are neutral does persist in the community. One needs only to look to the growing number of fitness landscape studies that focus on amino acid replacements, rather than nucleotide substitutions, as evidence. We have cited two recent examples from this literature to support this statement and, additionally, make reference to a third that took a fitness landscape approach at the nucleotide level. Our penultimate paragraph now begins with, “Despite mounting evidence to the contrary, the assumption that synonymous mutations are neutral remains deeply embedded in genetics. One need only look as far as the growing number of fitness landscape studies that focus on amino acid replacements which, by definition, involve nonsynonymous substitutions, the exception being a recent study in yeast showing that most synonymous mutations have small or negligible effects on fitness.”

5) Regarding the use of the codon adaptation index (CAI) as a metric of adaptation for codon usage. CAI is that it is a metric that looks for gene-level codon usage differences by comparing the distribution of codons used within the gene to a reference set. As the gene used in the study consists of ~300 codons, changing a single codon through a point mutation is highly unlikely to ever result in a significant change in CAI as it has little impact on the total codon distribution of the gene. The lack of association is therefore not surprising. As an alternative method for testing for codon usage adaptation, the authors could quantify the difference in ancestral and mutant codon usage for the specific mutated codon and compare that difference to the fitness benefit shown by that isolate. If adaptation is driven by codon usage, one would expect a significant positive correlation between the gain in codon preference at the mutated site and fitness. The use of this or any similar metric that specifically focuses on the mutated codons would substantially improve the work. Note that these additional experiments would strengthen the paper but are not required. Nevertheless, the limitation of the CAI metric used should be taken into account.

It is not entirely clear to us what test the reviewers are suggesting we do here because, as they point out, the change in CAI for any given nucleotide mutation is extremely modest. Perhaps we were unclear in what we wrote. We have revised our wording to reflect that the test of the codon adaptation hypothesis examines the change in CAI as a result of the mutation introduced at that codon, relative to the CAI of the WT, against the fitness difference between the mutant and WT estimated experimentally. We find no relationship, perhaps not surprisingly. We also added a statement that underscores the modest changes in CAI expected, in line with the reviewers’ comments. The new text now reads: “We could find no evidence for either explanation in our sample: a regression of fitness on distance from the start codon of *gtsB* was not statistically significant (permutation of residuals, P = 0.20), nor was there a significant relationship between change in fitness and change in codon adaptation index (CAI; P = 0. 40) or tRNA adaptation index (tAI; P = 0.53), both measures of the degree of codon usage bias. This is perhaps unsurprising given that CAI is a gene-level codon usage metric, and a change in a single codon is unlikely to have a large effect on the overall CAI value for either the whole gene or a portion thereof.”

6) One piece of information that may not have been taken full advantage of is the difference between the number of sites targeted for mutagenesis (112) and the number recovered (34). Assuming that the authors sampled a large number of clones from the transformed library (say 1000+), the loss of sites from the sampled mutants means that mutation at these sites is lethal. This could add significant new information into the deleterious region of the DFE. If a saturating number of clones was not sampled, it should be made clear in the Materials and methods.

Respectfully, we think the reviewers are confusing the number of sites with the number of mutations. We recovered 110 point mutants from the 112 sites targeted for mutagenesis. The loss of strains is normal with regards to the protocol's inherent failure rate.

7) The phylogenetic analyses showing that the 38-3T has arisen independently should be detailed because the correlation shown relies on the fact that these mutations are independent. If the points shown on Figure 4 are not independent, i.e. some strains share the mutations because of common ancestry and not because it was acquired, the correlation could derive from other factors also correlated with this ancestry.

This is, in fact, precisely what we did, as explained in the following text: “For each site in *gtsB*, the number of evolutionary events was calculated by comparing the inferred state at the beginning and end of each branch and counting the number of transitions.” To improve clarity, we have added, "… our highest fitness mutation, 39-3T, which is synonymous, arises independently multiple times across the phylogeny, even when common ancestry is taken into consideration."